# Developing a comprehensive model of home-based formal care for elderly adults in Iran: A study protocol

**Khorshid Mobasseri[1,2], Ahmad Kousha[3], Hamid Allahverdipour[3], Hossein Matlabi[2,4]** *

**1** Student Research Committee, Tabriz University of Medical Sciences, Tabriz, Iran, **2** Department of Geriatric Health, Faculty of Health Sciences, Tabriz University of Medical Sciences, Tabriz, Iran,
**3** Department of Health Education and Promotion, Tabriz University of Medical Sciences, Tabriz, Iran,
**4** Research Center for Integrative Medicine in Aging, Tabriz University of Medical Sciences, Tabriz, Iran

* hm1349@gmail.com

## Abstract

### Background

Due to the increasing Iran's aging population, designing a home care model is necessary. However, the data on designing the home care model for elderly adults among developing countries are limited. This study will be carried out to develop a formal home-based care model for elderly adults in Iran.

### Methods

This multi-method study will include three phases: First, Mixed-methods sequential explanatory study including two steps: One, survey to determine the prevalence of dependence on formal and informal caregivers among people aged ≥ 60 years living in Tabriz metropolis; and two, content analysis approach includes face-to-face, semi-structured interviews with the older adults receiving formal care at home, their caregivers and relevant key informants on the characteristics of care and caregiver, challenges and expectations of standard care. In phase 2, a scoping review will be used to find out the components of home care in other countries, such as care provider organization, caregivers training, and financing. PubMed, Scopus, Web of Science, EMBASE, Google scholar databases and grey literature will be run to retrieve relevant evidence using proper MeSH terms. In phase3, the triangulation method (using the results of the previous phases, reviewing national upstream documents and the focus group discussion) will be done to reach consensus and design the initial model for the Iranian context. In the following, a Delphi study will be conducted on the validation and feasibility of the developed model.

### Discussion

The current health infrastructure in Iran is focused on caring for younger people, despite the near future population aging. Most studies have addressed the challenges of geriatric care, but no study has addressed the various dimensions of home care in Iran and how to provide

**Data Availability Statement:** No datasets were generated or analysed during the current study. All relevant data from this study will be made available upon study completion.

**Funding:** This study was part of KHM's Ph.D. dissertation supervised by AK approved and supported financially by Tabriz University of Medical Sciences, Tabriz, Iran. The grant number is (68225). The funders had no role in study design, data collection and analysis, decision to publish, or preparation of the manuscript.

**Competing interests:** NO authors have competing interests.

**Abbreviations:** ADL, Activities of Daily Living; CwC, Caring with Confidence; IADL, Instrumental Activities of Daily; LTC, long-term care; SD, standard deviation; SPSS, Statistical Package for the Social Sciences; TRACK, Training and recognition of informal Carers' Skills; WHO, World Health Organization.

this service to elderly adults. Providing a comprehensive model of home care for the elderly can improve the quality of life and satisfaction of the elderly and their caregivers.

## 1. Background

One of the most social developments of the 21st century is the population aging. According to the United Nations, older adults are considered persons aged 60 years or over [1]. According to the latest census in 2016, the population aged 60 and over in Iran was more than 7 million people (9.3% of the total population), which is estimated to reach 8 million 849 thousand people (9.6% of the total population) in 2030 [2]. Increasing life expectancy and the number of elderly adults with chronic diseases lead to long-term care (LTC) needs. Therefore, this issue will be one of the major challenges of health care systems in most countries [3]. According to the National Institute on Aging, long-term care includes various services that address individual health and personal needs in two types, formal and informal care. The purpose of LTC is to maintain independence and security of the person [4]. Informal caregivers are usually family members or friends who care for elderly adults, typically without payment [5]. Family caregivers are considered the main resource of the health care system and the most cost-effective way to reduce financial burden of care and hospitalization [6]. Informal caregivers often experience various problems, including health decline [7], labor market problems and poverty [8], learning needs [9], and care burden [10]. Furthermore, women, especially daughters, who account for the main family caregivers, are expected to reduce their capacity to work at home due to the increasing their participation in the labor market and changing the family structure from extended to nuclear [11]. In response to population aging, personal preferences, reducing the burden on family caregivers and policies on encouraging aging in place, many countries have promoted formal HC while supporting informal caregivers and prioritizing providing care by relatives [12–14]. Formal care is provided by paid caregivers or healthcare institutions and includes care at homes, hospitals, nursing homes, and other types of care centers [15]. Basic or non-specialist formal care includes Activities of Daily Living (ADL) and Instrumental Activities of Daily Living (IADL), such as personal hygiene, dressing, eating, transportation and medication management. Home-based health care is provided by professionals such as educated nurses or health workers and includes nursing and medical services [16].

Formal home care that has all the components of structured care, like governance, trained caregivers along with involving the trained family caregivers, sustainable financing, providing benefits based on needs assessment and evaluation of care process, will improve older adults' quality of life and reduce health care costs [14].

Despite the population aging in Iran, the current health infrastructure is focused on caring for young people [17]. Policies have been developed to support providing community-based care for older adults in Iran, such as strategic planning of Iran's National Document for older adults and General population policies in Iran. Nevertheless, in practice, these plans and policies are not implemented or implemented incompletely and negatively [18]. There are many trustees for older adults' affairs, such as the Ministry of Health, the Imam Khomeini Relief Committee, and the Social Security Organization, but these trustees conflict with each other [19]. There is no organized home care, and home care is not provided by trained and qualified caregivers with high-quality standards [20]. However, there are institutions such as the Nursing Consulting Institute in Iran, which, in addition to their other activities, also provide home care services to older adults living in the community [21]. At present, a study that shows who provides home care to older adults and what factors determine dependency in them in Tabriz

city was not found. Unqualified paid caregivers and nurses also face challenges in care, but no study explores the experiences and challenges of all stakeholders involved in geriatric home care. Therefore, it is necessary to discover the status of home care in Iran and the challenges in this regard. After clarifying these issues, in the following, by using the experiences of the leading countries and the experiences of experts, the model of formal home care for elderly adults can be developed and finally validated. Thus, due to the rapid demographic changes, older people's special and growing needs, care burden of informal caregivers and high cost of hospitalization, reforming and redesigning the home- based long-term care system for Iranian older people is required. The specific importance of this study is to address the following hypothesis:

*Providing home care for Iranian older adults requires a comprehensive model.*
Objectives:

1. Assessing the status of providing home care services for older adults in Iran

2. A review of the structure of home care for elderly adults in different countries

3. Developing a comprehensive model for providing home-based long-term care for Iranian elderly adults and validation and feasibility of the model for the context of Iran

## 2. Materials and methods

This study protocol has undergone peer-review and approved by the Research Ethics Committee in Tabriz University of Medical Sciences (IR.TBZMED.REC.1400.934). The multi-method study will be conducted in four stages as follow (Fig 1).

### 2.1 Phase 1

The first phase is a mixed-methods sequential explanatory design which consists of 2 steps.

**2.1.1. Step one.** A quantitative cross-sectional study will be conducted to determine dependency level of elderly adults in Tabriz city on ADL and IADL, home-based supportive and health care services provided, characteristics of the caregivers and influencing factors on disability (ADL and IADL) in Iran. According to the Nagi's conceptual framework for studying disability, interaction between diseases and functional limitations leads to disability [22]. To unify the assessment of disability level, The World Health Organization's International Classification of Functioning, Disability, and Health (ICF) describes functional disability as a difficulty in executing activities of daily living (ADLs) and instrumental activities of daily living (IADLs) independently [23]. These activities are important because even the need for help in doing them makes a person unable to fulfill his social roles properly [24]. Similar to previous studies, functional disability will be measured as needing help with, or dependency to perform, at least one of the ADL/IADL activities [25, 26].

Therefore, the research hypothesis in this step can be stated as follows:

*There is an association between underlying chronic diseases and difficulty in performing ADL and IADL.*

*2.1.1.1. Sampling.* The Cochran's Sample Size Formula was used to calculate the number of people who will enter the study.

$$N_0 = \frac{Z^2 pq}{e^2}$$

The study used stratified- cluster sampling design. Simple random sampling across the Tabriz city could have been utilized to recruit elderly adults. However, using the simple random sampling, a large enough older people were needed to be representative sample. The

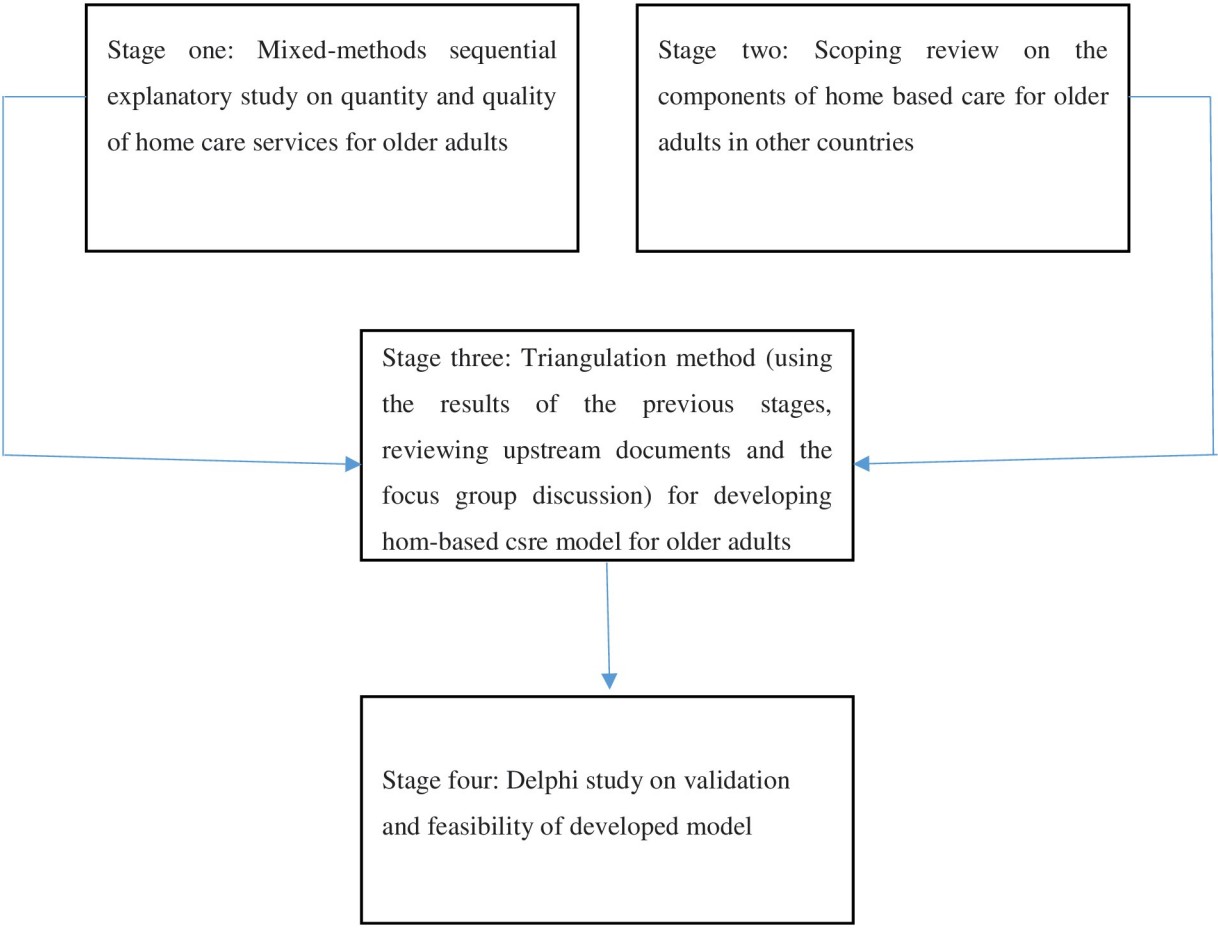

**Fig 1. The study flowchart.**

population of Tabriz is geographically diverse. Therefore, random sampling of elderly adults across the city would have been impractical and expensive. Stratified- cluster sampling is effective in reducing sampling error [27, 28]. A stratified cluster sampling approach was therefore used to ensure the selected sample was representative of the city.

The Tabriz city, as one of the metropolises of Iran, according to the latest population and housing census in 2016, has a population of 1.5 million [29]. Areas of Tabriz city will be selected as clusters and 55 comprehensive urban health centers will be selected as stratifies. The inclusion criteria for this study will be age ≥60 years and willingness to participate in the study. We will also check the cognitive status of participants using the valid Persian version of Abbreviated Mental Test Score (AMT) before filling out the questionnaire. If elderly adults have a cognitive disorder, the person closest to the older person will be asked questions as the representative. Normal cognitive status in the AMT questionnaire is considered > 7 points. The validity and reliability of the Persian version of the scale have been confirmed on patients with bipolar disorders, with the Cronbach's alpha coefficient of 0.76 [30]. The exclusion criteria will be to refuse participation in the study. Considering the lack of a similar study on the proportion of the elderly adults receiving supportive and health care at home in similar countries with Iranian context, the "p" in the formula was considered 50%. The sample size is calculated to be 385 people, considering the cluster sampling and the dispersion within the clusters,

taking into account the cluster effect (1.5 = design effect) and adding 10% to prevent the drop of samples, eventually 700 older people aged 60 years and older will enter this step. Because the questions of the questionnaire will be asked in the first step of phase1 by phone, therefore, verbal consent will be received from the respondents.

*2.1.1.2. The instrument (Home care services for elderly adults).* The instrument will consist of two sections: The first section of the instrument will include information about the socio-demographic characteristics of the participants and questions about receiving health-related services at home. Services mentioned in previous evidence will be used to determine health-related services content [31, 32]. Questions on the characteristics of care, caregivers, and paid caregivers will also be included. Questions on the attributes of care, such as the number of hours of care and the frequency of care, will be assessed in this section. The second section of the instrument will consist of type of supportive service received in terms of ADL and IADL. In designing this part, the valid and reliable Persian KATZ questionnaire will be used to determine receiving care in ADL and IADL activities [33, 34].

After designing the instrument, the content and apparent validity will be conducted. For this aim, at least 15 experts with experience and knowledge of the subject will be invited to comment on each element. This will be revised based on feedback received. Following confirmation of the final instrument, it will be used to achieve the goal of this step.

*2.1.1.3. Statistical analysis.* Statistical analysis will perform using the Statistical Package for Social Sciences (SPSS 24.0, SPSS Inc., and Chicago, USA). Descriptive statistics on demographic and medical characteristics will performed with mean, standard deviation (SD) and number (N), percentage (%) as applicable. The variables will be compared between participants with dependent, needs assistance and dependent in ADL and IADL, using the chi-square and Fisher's exact test (for small-sized samples). Also, simple (Univariate) and multiple linear regression analyses will be utilized to assess the effect of different variables on ADL and IADL by utilizing the total score of ADL and IADL as dependent variables. P value of less than 0.05 will be considered as statistical significance.

**2.1.2. Step two.** *2.1.2.1. Study design.* An exploratory qualitative approach will be used for this step to achieve a more in-depth understanding of participants' experiences [35]. We will conduct qualitative interviews with stakeholders to express their experiences about the status and existing challenges of providing home care for elderly adults.

*2.1.2.2. Sampling and data collection.* In Iran, any institution related to care must be licensed under certain regulations. The Ministry of Health announces rules for establishing facility, standards, workforce, tariffs and quality monitoring checklists. Deputy Chancellor for Treatment is responsible for monitoring the quality. One of these centers is the counseling and nursing care center, which provides home-based care [21]. We will explore the experiences of various participants who have extensive experience with home care issues and challenges for elderly adults. The study participants will be selected through purposeful and snowball sampling method. All participants will complete a consent form before entry into this phase. Confidentiality of the participants' information will be considered.

The first group will consist of elderly adults or their close family members who spend the most time in care. The second group will be non-qualified paid caregivers employed by nursing consultation center with at least five years of work experience in domestic care, including (Instrumental) Activities of Daily Living. The third group will be skilled caregivers (nurses) employed in home nursing with at least five years' experience in geriatric care. The fourth group will be heads of nursing consultation centers that hire nurses and caregivers for home care. The inclusion criteria for recruiting these participants will be to have a minimum of five years of experience working with caregivers. The fifth group will include faculty members or researchers in geriatric field. The experts who have at least two years of work or research

experience in geriatrics will be included in this step. Reluctance to participate in the study will be the exclusion criteria for all participants.

Data will be collected through semi-structured and in-depth interviews using guide comprising probing questions (Table 1). Each interview guide will consist of several main questions for each group of participants. All researchers and a number of geriatric experts will assess the validity of the interview guides. Issues and challenges related to home care will be major questions. Unwillingness to participate in study will be the exclusion criteria for all participants. Prior to the start of the interview, participants will be given a brief explanation of the purpose, the data collection process and the role of the researchers and study participants.

**Table 1. Interview guide for each group of stakeholders.**

| Group | Topics |
|---|---|
| **Older adults and their family members** | • Why do you use home care services?<br>• How did you get access to home care services? What challenges have you had in this regard?<br>• How do you pay for it?<br>• What are the characteristics of someone who serves you at home? (e.g., age, sex, educational level)<br>• In your opinion, what educational needs should be met if the family is to be the main caregiver of older adults in the future?<br>• What are your expectations for the outputs of home care? |
| **Non-qualified caregivers and nurses** | • What are the reasons that have made you care for older adults at home?<br>• What is your feelings about providing care for older adults?<br>• What training have you seen in this field? Do you have a suggestion to improve the education of caregivers?<br>• What challenges, problems and issues do you have in providing care to the older adults?<br>• What are your expected outcomes of care?<br>• What are your suggestions for improving the quantity and quality of home care services?<br>• Are you satisfied with your job? What are your expectations for your career advancement? |
| **Heads of care institutions** | • Why do you work in the field of home care for older adults?<br>• How is staff training done in your institution? Do you have any suggestions in this regard?<br>• How is the competence of caregivers determined in this institution? Do you have any suggestions to improve this?<br>• What challenges, problems and issues do you have in providing care to older adults? How to overcome these challenges?<br>• How do you pay the caregiver? What is your suggested solution to deal with financial challenges?<br>• How do you measure the quality of care? How can the quality of care be improved? |
| **Experts** | • How would you describe the current home care system for older adults?<br>• Do you think it is possible to establish a home care system in Iran formally? Mention the reasons.<br>• What are the obstacles in this path?<br>• If we consider home care for older adults as a system, what components does this system include?<br>• What are the solutions for financing the context of Iran? How to cover the cost of long-term care?<br>• What can support be done for the workforce? |

Each interview will be held in the participants' preferred location. The interviews will continue until reaching data saturation point when no new information is obtained [36]. The conversations will be digitally recorded and transcribed verbatim at the end of each interview session. Each interview will be conducted in a preferred place for participants.

*2.1.2.3. Data analysis.* An inductive content analysis approach will be used to analyze transcribed verbatim interviews to identify key concepts. content analysis is appropriate approach for analyzing data from a qualitative study that explores an unknown phenomenon, and there are limited or no studies on the research question [37]. Furthermore, a well-suited method for extracting codes, categories and themes based on the determination of trends, patterns and frequency of used words is content analysis [38]. Data analysis will be conducted during data collection process. Transcriptions will read frequently to obtain a general understanding of participants' statements in the line with the objectives. This will lead to extraction of meaning units or initial codes. Extracted codes will be merged and categorized in groups based on similarities and differences. In the following, categories will be classified in subthemes. Final subthemes and their relationship with each other, will be reviewed in order to reach consensus regarding the unite themes emerging from data. Themes and concepts will be finalized according to suggestions of research team members. MAXQDA 12, as a qualitative data analysis software, will be used to index references and annotate in margin beside the text.

*2.1.2.4. Trustworthiness.* To increase the validity and reliability of qualitative research, we will use the criteria that Lincoln and Guba introduced [39]. To increase the credibility of the findings, we will include key informants with sufficient experience in the field of aging and geriatric. Checking all data by two members of the research team will enhance the conformability of our study. In order to increase the dependency, we will document all steps of the study accurately. In addition, participants with maximum variation will be recruited to enhance the transferability of the findings.

## 2.2. Phase 2

This scoping review study will be conducted according to the framework proposed by Arkesy and O'Malley developed in 2005, including six steps [40]:

**2.2.1. Step one: Identification of the research question.** This step will aim to identify how home-based long-term care for elderly adults in different countries by searching in related databases.

**2.2.2. Step two: Identification of the relevant evidence to the research question.** Information sources will be carried out in the following resources: PubMed, Scopus, Web of Science, EMBASE, Google scholar and other sources of information to identify grey literature. All found articles will be exported to Endnote X8 software. After removing duplicates, titles and abstracts of articles will be reviewed based on inclusion and exclusion criteria and, if eligible, will be included in the study.

• Inclusion criteria:

Relevance of articles, reports, websites on the components of home care (trustee organization, staff training process, financing of long-term care services, issues related to long-term insurance, legislation, expected results of long-term care and evaluation need to receive care, quality assessment and care outputs. We will not limit the selection of countries because otherwise we may lose the information of some countries. With a systematic search, any country that has information on the structure of home care will be found, and we will include the information from these countries in this study.

• Exclusion criteria:

Lack of information about the components of home care, non-English language, nontent and documentation of invalid sources (eg, websites, blogs, online magazines that are not reviewed by the reviewer, lack of authors' names and logos), experimental studies.

**2.2.3. Step three: Study selection.** Two members of the research team independently will screen all titles, abstracts, and full texts. After finalizing the documents, the required information will be extracted. PRISMA tool will be used to evaluate the quality of studies [41, 42]. An example of a search strategy in the PubMed database is provided in Table 2.

**2.2.4. Step four: Data categorization.** Data classification will be carried out based on the information in the documents and articles. Summarizing and reporting the results. Information such as type of study, subject of study, country and publication year will also be summarized and categorized.

**2.2.5. Step five: Conclusion, summation and reporting results.** Due to the wide scope of our research question we will contain this final stage to a content analysis where we will immerse in the results of articles, identify, extract the main areas, and organize results into specific categories.

## 2.3. Phase 3

In the third stage, the research team will develop the initial model using the triangulation method (using the results of the previous stages, reviewing upstream documents and the expert panel) and assessing validation and feasibility of the developed model by the Delphi study. All upstream documents related to the health of elderly adults, including general health policies, Iran Vision 1404 document, Iran five-year development plans, national document on aging and other documents related to the elderly adults will be extracted. The data collection tool for this section is a data extraction form that will be designed to provide information. The purpose of designing this form is to create a regular schedule and prevent data missing. This form can help to classify and organize the data obtained from different documents in a structured and systematic way and give a comprehensive view to the reader [43]. The information on title, type (Law, policy, regulation or other types of documents), the principle, paragraph or article of the document, publication date and place, stakeholders and content of the document will be included in the form.

Identification and access to documents through scientific databases, electronic portals and Internet sites related to related organizations (Islamic Consultative Assembly, Ministry of Health, Ministry of Welfare, State of Welfare Organization and Imam Khomeini Relief Committee, World Health Organization (WHO), etc.) will be searched. If necessary, through face-to-face visits to the related organizations, the required data will be collected and extracted.

The initial model will be designed using the results obtained from the previous steps. These sessions will continue using the pre-set instructions until we reach the consensus. The focus group discussion session will consists of a group of care-related experts and key informants, managers and policymakers in the geriatric care field. The selection of experts will be done using purposeful sampling and, if necessary, snowball sampling [44]. Informed consent will be obtained from the participants before the sessions begin. The most important factor in determining the number of focused group discussion sessions is reaching saturation [45]. The number of sessions will continue until data saturation is reached in the present study. So that no new point of view is added to the discussion and the model. The facilitator/interviewer should be familiar with the discussion, have previous experience, and at the same time play a neutral role in facilitating group communication, leading the discussion, encouraging participants to discuss, and collaborating in the discussion [46]. The selection criteria will be as follows: having expertise in the geriatric field, or related disciplines, having at least three years of work

**Table 2. An example of search strategy in the PubMed database is provided below.**

| No | Search Query | search Results (Number of Studies) |
|---|---|---|
| 1 | (("old"[Title/Abstract] OR "older"[Title/Abstract] OR "elderly"[Title/Abstract] OR "senior"[Title/Abstract] OR "Aged"[Title/Abstract] OR "geriatric*"[Title/Abstract] OR "vulnerable group"[Title/Abstract]) AND ("Cost"[Title/Abstract] OR "Funding"[Title/Abstract] OR "Insurance"[Title/Abstract] OR "organization*"[Title/Abstract] OR "institute*"[Title/Abstract] OR "Training"[Title/Abstract] OR "skill*"[Title/Abstract] OR "Worker*"[Title/Abstract] OR "Staff"[Title/Abstract] OR "care provider*"[Title/Abstract] OR "quality indicator*"[Title/Abstract] OR "Policy"[Title/Abstract] OR "Guideline*"[Title/Abstract])) AND (((Long-term care[Title/Abstract]) OR ("Formal care"[Title/Abstract] OR "informal care"[Title/Abstract])) AND ("Home care"[Title/Abstract] OR "home-based services"[Title/Abstract] OR "Social home care"[Title/Abstract] OR "Home healthcare"[Title/Abstract] OR "Community Health Service*"[Title/Abstract])) | 493 |
| 2 | ("old"[Title/Abstract] OR "older"[Title/Abstract] OR "elderly"[Title/Abstract] OR "senior"[Title/Abstract] OR "Aged"[Title/Abstract] OR "geriatric*"[Title/Abstract] OR "vulnerable group"[Title/Abstract]) AND ("Cost"[Title/Abstract] OR "Funding"[Title/Abstract] OR "Insurance"[Title/Abstract] OR "organization*"[Title/Abstract] OR "institute*"[Title/Abstract] OR "Training"[Title/Abstract] OR "skill*"[Title/Abstract] OR "Worker*"[Title/Abstract] OR "Staff"[Title/Abstract] OR "care provider*"[Title/Abstract] OR "quality indicator*"[Title/Abstract] OR "Policy"[Title/Abstract] OR "Guideline*"[Title/Abstract]) | 277481 |
| 3 | ((Long-term care[Title/Abstract]) OR ("Formal care"[Title/Abstract] OR "informal care"[Title/Abstract])) AND ("Home care"[Title/Abstract] OR "home-based services"[Title/Abstract] OR "Social home care"[Title/Abstract] OR "Home healthcare"[Title/Abstract] OR "Community Health Service*"[Title/Abstract]) | 1660 |
| 4 | "Home care"[Title/Abstract] OR "home-based services"[Title/Abstract] OR "Social home care"[Title/Abstract] OR "Home healthcare"[Title/Abstract] OR "Community Health Service*"[Title/Abstract] | 23704 |
| 5 | (Long-term care[Title/Abstract]) OR ("Formal care"[Title/Abstract] OR "informal care"[Title/Abstract]) | 26284 |
| 6 | (Long-term care[Title/Abstract]) AND ((("Home care"[Title/Abstract] OR "home-based services"[Title/Abstract] OR "Social home care"[Title/Abstract] OR "Home healthcare"[Title/Abstract] OR "Formal care"[Title/Abstract] OR "informal care"[Title/Abstract] OR "Community Health Services"[Title/Abstract]) AND (old[Title/Abstract] OR older[Title/Abstract] OR elderly[Title/Abstract] OR senior[Title/Abstract] OR Aged[Title/Abstract] OR geriatrics[Title/Abstract] OR vulnerable group[Title/Abstract])) AND (Cost[Title/Abstract] OR Funding[Title/Abstract] OR Insurance[Title/Abstract] OR organization[Title/Abstract] OR institute[Title/Abstract] OR Training[Title/Abstract] OR skill[Title/Abstract] OR Workers[Title/Abstract] OR Staff[Title/Abstract] OR Care providers[Title/Abstract] OR Quality indicators[Title/Abstract] OR Policy[Title/Abstract] OR Guideline[Title/Abstract])) | 499 |
| 7 | (("Home care"[Title/Abstract] OR "home-based services"[Title/Abstract] OR "Social home care"[Title/Abstract] OR "Home healthcare"[Title/Abstract] OR "Formal care"[Title/Abstract] OR "informal care"[Title/Abstract] OR "Community Health Services"[Title/Abstract]) AND (old[Title/Abstract] OR older[Title/Abstract] OR elderly[Title/Abstract] OR senior[Title/Abstract] OR Aged[Title/Abstract] OR geriatrics[Title/Abstract] OR vulnerable group[Title/Abstract])) AND (Cost[Title/Abstract] OR Funding[Title/Abstract] OR Insurance[Title/Abstract] OR organization[Title/Abstract] OR institute[Title/Abstract] OR Training[Title/Abstract] OR skill[Title/Abstract] OR Workers[Title/Abstract] OR Staff[Title/Abstract] OR Care providers[Title/Abstract] OR Quality indicators[Title/Abstract] OR Policy[Title/Abstract] OR Guideline[Title/Abstract]) | 2831 |

(*Continued*)

**Table 2.** (Continued)

| No | Search Query | search Results (Number of Studies) |
|---|---|---|
| **8** | (("Home care"[Title] OR "home-based services"[Title] OR "Social home care"[Title] OR "Home healthcare"[Title] OR "Formal care"[Title] OR "informal care"[Title] OR "Community Health Services"[Title]) AND (old[Title] OR older [Title] OR elderly[Title] OR senior[Title] OR Aged[Title] OR geriatrics[Title] OR vulnerable group[Title])) AND (Cost[Title/Abstract] OR Funding[Title/Abstract] OR Insurance[Title/Abstract] OR organization[Title/Abstract] OR institute[Title/Abstract] OR Training[Title/Abstract] OR skill[Title/Abstract] OR Workers [Title/Abstract] OR Staff[Title/Abstract] OR Care providers[Title/Abstract] OR Quality indicators[Title/Abstract] OR Policy[Title/Abstract] OR Guideline[Title/Abstract]) | 344 |

experience in the relevant field, having at least a master's degree or higher, trustees and providers of home care services, people engaged in research in a related field. If you do not want to continue, participants will allowed to leave any part of the study. Validation and feasibility of the developed model in the Delphi platform appears as a method to provide guarantees to use of the new model in the particular context [47]. Delphi technique is a structured process that is used to reach a consensus between experts and specialists. In other words, it is a structured process for collecting and classifying the knowledge of experts, which is done by distributing questionnaires among them and receiving feedback on the answers and comments [48]. For this purpose, first the Delphi questionnaire will be designed using the Likert scale based on components of the initial model Sampling used in the Delphi technique is based on purpose, and if the specialists are not identified, snowball sampling is also used [49]. Sampling used in the Delphi technique will be purposive and snowball sampling. The number of participants is usually less than 50 and mostly 15 to 20 people [50]. In this study, considering losing some participants, 25 experts from all over the country will be invited to attend the sessions. Another component of Delphi is the repetition of Delphi rounds with the aim of clarify the information acquired in the prior rounds access to new insights of the participants, controlled feedback and 70% consensus[51–53]. Although the number of rounds increases the accuracy of the information, it often causes fatigue and does not lead to effective results [54]. Therefore, in this study, between 1 and 3 rounds will be performed depending on the consensus of opinions. The inclusion criteria for the study's participants in the Delphi study will be the same as the criteria for the expert panel. Obtaining informed consent from the participants will also be done at this stage.

Consensus is measured using various measures such as percentage and mean scores [55]. In this questionnaire, 12 items will be examined using similar studies, and the mean score given to each item will be used to confirm the validity of the developed model. So that the scale will be from 1 to 9. $\geq 6$ item confirmation, $6>$ and $\geq 4$ need to be raised in the second round of Delphi and $4>$ rejection (105).

## 3. Discussion

This is the first study to provide a formal care model for the elderly adults living at home in Iran. It is different from the previous studies that were conducted in Iran in this area. Several studies have compared elderly adults living in the community and nursing homes [56–58]. Others have addressed the characteristics of care and caregivers of Alzheimer's patients in nursing homes or living at home and the care burden of caregivers with Alzheimer's disease [59, 60]. Some have also identified the problems of caregivers [61–63]. Several studies are

about the burden of disability in Iran; for example, in 2012 in Iran, 13.2% of women and 12.6% of men were dependent for activities of daily living [64]. In another study in 2011, the prevalence of disability was reported 11% among the elderly adults of Tehran [65]. Previous studies have shown that dependence increases with age [66, 67]. Furthermore, there are studies in Iran that have been conducted with the aim of investigating the relationship between different variables and instrumental/non-instrumental daily living activities [68–70]. There is no study on the prevalence of dependence in daily and instrumental activities and health-related issues of the elderly adults in Tabriz. We will consider ADL and IADL as dependent variables and assess the effect of different variables on them.

Formal home care services in European countries have helped a quickly rising proportion of elderly adults [71]. Taking into account that Iran is the second country to experience the fastest increase in the aging population between 2015 and 2050, there is a need for supportive strategies to maintain their social and economic security [72]. In studies that address the challenges of care for elderly adults in Iran, one of the solutions to compensate for the shortage of geriatricians and inadequate access to these services is to train skilled home care workers [17, 73].

In a qualitative study that explores the professional experiences of home care nurses about the concept of professionalism, three main categories were identified from their perspective, including attention to basic values, social capital, and maintaining quality and standards of care [74]. According to previous studies, caregivers face physical, mental and financial pressures and insufficient job support, which ultimately causes high caregiver turnover [75, 76]. Moreover, Home care in Iran is not in appropriate position due to problems in the educational system, cultural and security obstacles. Therefore, providing an proper infrastructure is a necessity in the context of Iran [77]. Getting information on care components in leading countries in geriatric home care services can make these services more effective and direct the resources into meeting the challenges of elderly adults [78]. The main common trends among Member countries of The Organization for Economic Cooperation and Development (OECD) include focusing on the elderly population with the greatest needs, decentralization, and encouraging private organizations to provide services and involve families in the care system [79]. Moreover, given the rapidly aging population of the world, it is expected that in the future, home care services should be person-centered, integrated and of high quality and in accordance with up-to-date standards and provided by specialists [80, 81]. Models have been designed for the home care process in Iran and other countries [82, 83], but a model that specifically focuses on the elderly population and addresses all domains of home care such as its management levels, human resources system, and financing was not found. Therefore, one of the aims of this study is to achieve the components of home care services and design the model for Iran.

According to the claim of the World Health Organization, policies should be formulated based on the needs of elderly adults in order to provide integrated services with a person-centered approach [84]. The financing system and its sustainability are among the most important key that have been addressed in advanced countries. In every country, depending on the country's resources, different methods are used for financing, such as insurance, municipal taxes, government subsidies, and out-of-pocket payments or mix of resource [85, 86]. Due to the sanctions, the oil selling, which was the main governmental financial source, is decreasing. Therefore, choosing complementary and flexible methods such as insurance and taxes can cover the costs of the health system [87, 88].

The role of the family in the care process as a trained auxiliary force and as a factor in reducing the burden of care will be very impressive [89]. Otherwise, with the increasing demand for long-term care and the lack of a formal labor force, trained formal and informal caregivers can complement each other [90–92]. Therefore, projects such as TRACK (Training

and recognition of informal Carers' Skills) in Europe and the Caring with Confidence (CwC) program to support informal caregivers have launched several training programs for skills in the field of self-care and care for elderly adults and provide them with a valid certificate [93].

Therefore, the results of this study will provide a comprehensive approach to taking care of elderly adults at home and use the experiences of leading countries to extract the main components of organized care by a multidisciplinary and skilled workforce with the cooperation of the family.

## 4. Conclusion

Success in effective planning in the geriatric field requires a comprehensive study of domains of home care in Iran and successful countries. Therefore, the results of this study can provide valuable information to policy makers to provide an evidence-based home care for Iranian elderly population. The experiences of successful countries can also be helpful in accurate policy making, sustainable financing, strengthening the training system of caregivers and the service delivery process. The model that will be designed will probably be a solution for the growing needs of Iran's elderly population in the near future, improving their quality of life and their caregivers, and increasing satisfaction level among all involved stakeholders.

## Supporting information

**S1 File. SPIRIT 2013 checklist: Recommended items to address in a clinical trial protocol and related documents.**
(DOC)

## Author Contributions

**Conceptualization:** Khorshid Mobasseri, Ahmad Kousha, Hamid Allahverdipour, Hossein Matlabi.

**Methodology:** Khorshid Mobasseri, Ahmad Kousha, Hamid Allahverdipour.

**Supervision:** Ahmad Kousha, Hamid Allahverdipour, Hossein Matlabi.

**Writing – original draft:** Khorshid Mobasseri, Hossein Matlabi.

**Writing – review & editing:** Ahmad Kousha, Hamid Allahverdipour, Hossein Matlabi.

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
