## [Decision Letter · Decision Letter 0]

20 Feb 2023

PONE-D-22-33262Developing a Comprehensive Model of Home-Based Formal Care for Older Adults in Iran: A Study ProtocolPLOS ONE

Dear Dr. Matlabi,

Thank you for submitting your manuscript to PLOS ONE. After careful consideration, we feel that it has merit but does not fully meet PLOS ONE’s publication criteria as it currently stands. Therefore, we invite you to submit a revised version of the manuscript that addresses the points raised during the review process. Be sure also to:Provide elaboration on the justification of using content analysis in Phase 1, Step 2 Justify the exclusion of experimental studies in the scoping review protocolThe manuscript would benefit from some minor editingPlease submit your revised manuscript by Apr 06 2023 11:59PM. If you will need more time than this to complete your revisions, please reply to this message or contact the journal office at plosone@plos.org. Please include the following items when submitting your revised manuscript:

We look forward to receiving your revised manuscript.

Kind regards,

Easter Joury

Academic Editor

PLOS ONE

Journal Requirements:

“No, the funders had and will not have a role in study design, data collection and analysis, decision to publish, or preparation of the manuscript.”

5. Please include a copy of Table 3 which you refer to in your text on page 9.

Reviewers' comments:

Reviewer's Responses to Questions

**Comments to the Author**

1. Does the manuscript provide a valid rationale for the proposed study, with clearly identified and justified research questions?

Reviewer #1: Yes

2. Is the protocol technically sound and planned in a manner that will lead to a meaningful outcome and allow testing the stated hypotheses?

Reviewer #1: Yes

3. Is the methodology feasible and described in sufficient detail to allow the work to be replicable?

Reviewer #1: Yes

4. Have the authors described where all data underlying the findings will be made available when the study is complete?

Reviewer #1: No

5. Is the manuscript presented in an intelligible fashion and written in standard English?

Reviewer #1: Yes

6. Review Comments to the Author

You may also provide optional suggestions and comments to authors that they might find helpful in planning their study.

Reviewer #1: Dear authors,

It was a pleasure to read and evaluate this excellent article, and I'd like to start by pointing out that your work has a well-thought-out research methods and a specific goal of creating a comprehensive model of formal home care for older people in Iran.

The use of mixed-methods, including a quantitative survey and qualitative interviews, allows for a comprehensive examination of the issues related to home-based care in Iran. For this kind of study, it is suitable to apply a scoping review to determine the elements of home care in other nations, the triangulation approach to develop an initial model for the Iranian setting, and a Delphi study to confirm and evaluate the model's viability.

However, there are a few questions I have regarding your methods:

- What are the specific countries that you studied in order to extract the main components of organized care?

- Can you provide more information about the Delphi study, specifically the number of rounds used and the criteria for selecting experts?

- Can you describe the data extraction form that will be used in phase 3 and explain its purpose?

- How will you ensure that the participants in the Delphi study are representative of the population of older adults living in Iran?

- Can you explain how the sample size was calculated and why the stratified-cluster sampling method was chosen?

In order to assess the proposed model from a financial perspective, I also urge you to think about integrating a cost-benefit analysis or cost-effectiveness analysis in your research. Examining the related financial matters, potential funding sources, and healthcare system savings for the proposed model's adoption could be helpful as well.

I anticipate reading your study's final findings since I think it has the potential to significantly advance the field of home-based care for senior citizens in Iran.

Thank you for your hard work.

Best regards,

7. PLOS authors have the option to publish the peer review history of their article (what does this mean?). If published, this will include your full peer review and any attached files.

Reviewer #1: **Yes: **MHD bahaa Aldin Alhaffar

---

## [Author Response · Author response to Decision Letter 0]

24 Mar 2023

Comments from the Editors and Reviewers:

Editor’s comments

comment Response

Provide elaboration on the justification of using content analysis in Phase 1, Step 2 Thank you for this comment. We explained the reason for using a content analysis approach to analyze qualitative data (page 12, line 271-277).

Justify the exclusion of experimental studies in the scoping review protocol Thanks for your comment. The philosophy of conducting experimental studies is to test the effect of independent variable(s) on dependent variable(s) and considering that the aim of this scoping review is to achieve the structure of home care in different countries, therefore, experimental studies cannot answer the research question.

Please ensure that your manuscript meets PLOS ONE's style requirements, including those for file naming. The PLOS ONE style templates. Thank you. We revised the article according to PLOS ONE's style requirements.

. Your ethics statement should only appear in the Methods section of your manuscript. If your ethics statement is written in any section besides the Methods, please delete it from any other section. We removed the ethics statement from the other sections and added it to the method section (page 7, lines 154-159)

Please include a copy of Table 3 which you refer to in your text on page 9. Thank you. We have not mentioned Table 3 in the text because Table 3 does not exist.

Please review your reference list to ensure that it is complete and correct. If you have cited papers that have been retracted, please include the rationale for doing so in the manuscript text, or remove these references and replace them with relevant current references. Any changes to the reference list should be mentioned in the rebuttal letter that accompanies your revised manuscript. If you need to cite a retracted article, indicate the article’s retracted status in the References list and also include a citation and full reference for the retraction notice. Thank you for your comment. We reviewed the list of references for accuracy. Because this article has been changed compared to the first version, therefore, we used the references related to the updated text.

Reviewer #1

Have the authors described where all data underlying the findings will be made available when the study is complete? NO No datasets were generated or analysed during the current study. All relevant data from this study will be made available upon study completion.

What are the specific countries that you studied in order to extract the main components of organized care?

 Thank you for your valuable comment. We will not limit the selection of countries because otherwise we may lose the information of some countries. With a systematic search, any country that has information on the structure of home care will be found, and we will include the information from these countries in our study. However, not all countries have a structured home care system for older adults, and this type of structure is more common in advanced countries such as Germany, Japan, South Korea, Denmark, Sweden, the United States of America, England, Switzerland, and similar countries. We added the justifications to the main text. (page 13, lines 309-313)

Can you provide more information about the Delphi study, specifically the number of rounds used and the criteria for selecting experts? Thank you for this valuable comment. We provided and highlighted in the text additional information about the number of Delphi rounds and why these rounds should be. (page 16, lines 369-376)

Due to the fact that experts in the geriatric field will be included in the Delphi study, therefore, the inclusion criteria for the study’s participants in the Delphi study will be the same as the criteria in the expert panel. We have included this sentence in the text repeated (page 16, lines 381- 383, 386-388).

Can you describe the data extraction form that will be used in phase 3 and explain its purpose? Thank you for your comment. The purpose of designing this form is to create a regular schedule and prevent data missing. This form can help to classify and organize the data obtained from different documents in a structured and systematic way and give a comprehensive view to the reader. The information on title, type (Law, policy, regulation or other types of documents), the principle, paragraph or article of the document, publication date and place, stakeholders and content of the document will be included in the form. We added this information in the text (page 15, lines 340- 346).

How will you ensure that the participants in the Delphi study are representative of the population of older adults living in Iran? Thank you for closely attention to the matter. Due to the fact that we will include experts in the Delphi study who have sufficient work experience and knowledge in the geriatrics field, therefore, because these experts are closely involved with the older adults’ issues, they can be a proper representative of the population of older adults living in Iran.

Can you explain how the sample size was calculated and why the stratified-cluster sampling method was chosen?

 Thank you for your comment. The Cochran’s Sample Size Formula was used to calculate the number of people who will enter the study. We have already included the details of the sample size calculation in the text. We highlighted this information.

The study used stratified- cluster sampling design. Simple random sampling across the Tabriz city could have been utilized to recruit older adults. However, using the simple random sampling, a large enough older people were needed to be representative sample. The population of Tabriz is geographically diverse. Therefore, random sampling of older adults across the city would have been impractical and expensive. Stratified- cluster sampling is effective in reducing sampling error [1, 2]. A stratified cluster sampling approach was therefore used to ensure the selected sample was representative of the city. We added this justification to the text (page 8, lines 177-190; 201-206).

In order to assess the proposed model from a financial perspective, I also urge you to think about integrating a cost-benefit analysis or cost-effectiveness analysis in your research. Examining the related financial matters, potential funding sources, and healthcare system savings for the proposed model's adoption could be helpful as well. Thank you for this valuable comment. In this model, effective resource allocation mechanisms will be clarified. For this purpose, using the experiences of advanced countries as well as the results of interviews with experts, financial resources will be determined according to the context of Iran. We added some information about financing in the discussion section (page19, lines 443-450).

Reviewer #2

Please check the grammar errors in the manuscript. For example: The title: it should be Elderly adults, not older adults (replace it in all manuscript)

In abstract: grammar errors: “few studies are in the home care area in developing countries.” It should be: few studies are conducted to evaluate home-based care model in the developing countries. 

 Thank you for this comment. We checked the grammatical errors and corrected them. We replaced elderly adults with older adults in all manuscript.

In methodology: phase 2: what is the relevance of using the data base for searching while your study is not a systematic review. 

 Thank you for this comment. It has been emphasized that one of the main methods for finding the answer to a question in the scoping review is a comprehensive search among a set of databases and one type of scoping review is a systematic scoping review in which reliable databases should be used for searching [3]. In addition, reliable websites and gray literature can also be helpful and useful [4].

In abstract: the last paragraph is a conclusion not a discussion. Furthermore, there is no result paragraph in the abstract. 

 Thanks for your insights and guidance. We changed the discussion section in the abstract. 

Because this article is a protocol study and the results have not been obtained, therefore the abstract and the main text do not have the results section.

In the objectives: it is another study to make a review of the structure of home care for older adults. All data related to this part should be shredded and you may use it to prepare a separated systematic review for example. Thank you for your comment. In order to design a home care model, we need to extract different aspects of home care in a study and use its results in the development of model. Considering the nature of the scoping review, which is preferable to the systematic review in clarifying the key concepts and dimensions of a subject, we chose scoping review [5].

The validation of the model is part of the development and not a separated objective. Thanks for your comment. We added validation of the model with model development and removed Phase 4. (Page 15, line 335)

The background is too long, it should be summarized to important and related studies. Thank you. We removed additional information from the introduction section.

The study design is confused and not clear. It should be rebuilt. It is a paper and not a thesis. 

 Thanks for your comment. In order to provide a comprehensive model of home care for the Iranian elderly, it was necessary to design a comprehensive and well-considered method. The use of mixed-method, including a quantitative survey and qualitative interviews, clarifies all issues related to home-based care in Iran. In the following, it is appropriate to design a scoping review to determine the domains of home care in other countries, the triangulation approach to develop a model for the Iranian context, and a Delphi study to evaluate the model's feasibility and reliability.

No result, no conclusion in the manuscript. 

 Thank you for useful comment. We added conclusion section to the manuscript (page 20, line 461-469), but since this is a protocol study, there are currently no results to report in the article.

The discussion is very plain in comparison with the methodology and the supposed results. Thank you. We removed additional information from the introduction section and improved the discussion section by referring to related studies (page 18, lines 409-415, 423-428, 430-433; page 19, lines 436-439, 443-450).

1. Pu X, Gao G, Fan Y, Wang M. Parameter estimation in stratified cluster sampling under randomized response models for sensitive question survey. Plos one. 2016;11(2):e0148267.

2. Sedgwick P. Stratified cluster sampling. Bmj. 2013;347.

3. Peters MD, Godfrey CM, Khalil H, McInerney P, Parker D, Soares CB. Guidance for conducting systematic scoping reviews. JBI Evidence Implementation. 2015;13(3):141-6.

4. Munn Z, Peters MD, Stern C, Tufanaru C, McArthur A, Aromataris E. Systematic review or scoping review? Guidance for authors when choosing between a systematic or scoping review approach. BMC medical research methodology. 2018;18:1-7.

5. Tricco AC, Lillie E, Zarin W, O’brien K, Colquhoun H, Kastner M, et al. A scoping review on the conduct and reporting of scoping reviews. BMC medical research methodology. 2016;16:1-10.

---

## [Editor Report · Decision Letter 1]

3 Apr 2023

Developing a Comprehensive Model of Home-Based Formal Care for Elderly Adults in Iran: A Study Protocol

PONE-D-22-33262R1

Dear Dr. Matlabi,

We’re pleased to inform you that your manuscript has been judged scientifically suitable for publication and will be formally accepted for publication once it meets all outstanding technical requirements.

Kind regards,

Easter Joury

Academic Editor

PLOS ONE

Additional Editor Comments (optional):

The authors successfully addressed the comments given.

There are only two final minor comments:

With respect to excluding experimental studies, sometimes such studies include contextual information that might be related to the structure of home care.

Although the authors edited the manuscript it still needs some further editing.

---

## [Editor Report · Acceptance letter]

12 Apr 2023

PONE-D-22-33262R1 

Developing a Comprehensive Model of Home-Based Formal Care for Elderly Adults in Iran: A Study Protocol 

Dear Dr. Matlabi:

I'm pleased to inform you that your manuscript has been deemed suitable for publication in PLOS ONE. Congratulations! Your manuscript is now with our production department. 

Kind regards, 

on behalf of

Dr. Easter Joury 

Academic Editor

PLOS ONE